# Biomarker Changes in Oxygen Metabolism, Acid-Base Status, and Performance after the Off-Season in Well-Trained Cyclists

**DOI:** 10.3390/nu14183808

**Published:** 2022-09-15

**Authors:** Francisco Javier Martínez Noguera, Cristian Marín-Pagán, Linda H. Chung, Pedro E. Alcaraz

**Affiliations:** Research Center for High Performance Sport, Catholic University of Murcia (UCAM), Campus de los Jerónimos Nº 135, 30107 Murcia, Spain

**Keywords:** endurance exercise, detraining, physiology, gasometry, blood gases

## Abstract

During the off-season, cyclists reduce their volume and intensity of training in order to recover the body from the high workload during the competitive season. Some studies have examined the effects of the off-season on cardiovascular, metabolic, and performance levels but have not evaluated oxygen metabolism, acid-base status, and electrolytes in cyclists. Therefore, our main objective was to analyze these markers in the off-season period (8 weeks) via finger capillary blood gasometry in well-trained cyclists. We found an increase in oxygen saturation (sO_2_) and oxyhemoglobin (O_2_Hb) (*p* ≤ 0.05) and a decrease in fat oxidation at maximum fat oxidation (FatMax) (*p* ≤ 0.05). In addition, we observed a decreasing trend of VO_2_ in the ventilatory threshold 2 (VT2) and maximum oxygen consumption (VO_2MAX_) (*p* ≤ 0.06) after the off-season in well-trained cyclists. Negative correlations were found between the pre–post off-season differences in the VO_2_ at ΔFatMax and ΔHCO_3_^−^ (bicarbonate ion) and between power generated at the ΔeFTP (functional power threshold) and the ΔVO_2MAX_ with the pH (r ≥ −0.446; *p* ≤ 0.05). After the off-season period, well-trained cyclists had increased markers of oxygen metabolism, decreased fat oxidation at low exercise intensities, and decreased VO_2_ at the VT2 and VO_2MAX_. Relationships were found between changes in the ΔeFTP and VO_2MAX_ with changes in the pH and between the pH and HCO_3_^−^ with changes in La^^−^^.

## 1. Introduction

The annual training schedule of an elite or well-trained cyclist can be divided into three phases: (1) preparation, (2) competition, and (3) transition [1]. Following a strenuous period of competition, the cyclists’ training load is significantly reduced for 2–3 weeks in the subsequent transition period to ensure recovery [2,3]. However, long periods of >4 weeks of reduced training could lead to decrements in performance and adaptations achieved before the transition phase (off-season) [4].

Short-term detraining is defined as the partial or total loss of training-induced anatomical, physiological, and performance adaptations due to the reduction or cessation of training for less than 4 weeks, as well as the lack of a sufficient training stimulus [5]. During the transition phase (i.e., the off-season), a decrease in cardiorespiratory and metabolic variables has been found together with changes at the muscular level as a consequence of the decreased volume and intensity in highly trained individuals [5,6]. Additionally, short-term detraining (<4 weeks of an insufficient training stimulus) has been shown to decrease the maximal oxygen uptake (VO_2MAX_) by 4–14% in endurance-trained athletes [5,7] and lower blood volume by 5–9% in endurance runners [8,9]. Also, an increased pulse rate at submaximal [8] and maximal [9] intensities, reduced stroke volume [7], lower ventricular mass [10], and impaired pulmonary ventilatory function have been observed [11].

On the other hand, when the training stimulus is insufficient in the long term (>4 weeks), many changes occur in the body’s systems. Specifically, there can be a 6 to 20% decrease in the VO_2MAX_ [7,10,12,13,14] and a 14 to 17% reduction in systolic volume during upright endurance exercise [7,10], and an increase in the heart rate at submaximal and maximal (5%) intensities [6]. In addition, a decrease in the left ventricular end-diastolic dimension of trained individuals during upright endurance exercise was observed in parallel with the systolic volume during 8 weeks of training cessation [10]. At the ventilatory level, alterations have been observed after long-term training cessation with a decrease in maximal ventilatory volume and an increase in ventilatory equivalents during submaximal exercise [15,16].

Another negative effect of long-term detraining on performance is the decrease in time-to-exhaustion [17]. Metabolic changes, such as an increase in the respiratory exchange ratio (RER), have also been found, indicating a higher carbohydrate utilization at the same intensity in trained athletes [12,15]. In addition, increased blood lactate levels during submaximal exercise at the same absolute and relative intensity [12,18] and lactate threshold occurring at a lower percentage of the VO_2MAX_ [12,13] have been observed. Additionally, an 8.4% decrease in the arterial-venous oxygen difference [7] and oxidative enzyme activities, such as citrate synthase, β-hydroxyacyl-CoA dehydrogenase, malate dehydrogenase, and succinate dehydrogenase, have been shown by 25–40% [7,12,19] at the muscle level. These reductions in mitochondrial enzymes are related to the observed long-term reductions in the VO_2MAX_ and arterial-venous oxygen difference [7,14]. Also, long-term cessation of training may decrease mitochondrial ATP production [6] and produce a shift from the fibre type (FT), a to b, in endurance athletes [12].

The measurement of blood gas parameters is a useful tool at the basic science level to follow the changes in markers of oxygen metabolism produced by detraining. Furthermore, at a practical level, coaches or sports science professionals can orientate their training sessions in a targeted way based on the changes in these parameters mediated by detraining.

The aforementioned studies have mainly examined physiological, biochemical, and metabolic marker changes without exploring blood gas markers after short- and long-term detraining. Such studies are necessary to understand how changes in training variables can affect performance, physiological, biochemical, and metabolic markers, providing further knowledge to sports scientists. We hypothesized that 8 weeks of reduced load off-season training would produce changes in finger capillary blood gasometry, in line with changes observed in physiological, biochemical, and metabolic markers in well- trained cyclists.

Therefore, the main objective of this study was to evaluate the changes produced in finger capillary blood gas markers during an 8-week off-season period, where well-trained cyclists decreased their training volume and intensity. The secondary objective was to assess whether possible changes in blood gas markers could affect performance, physiological, and metabolic markers in well-trained cyclists.

## 2. Materials and Methods

Twenty healthy male, well-trained cyclists were recruited. All had to meet the following inclusion criteria: (a) aged 18–45 years; (b) BMI 19.0–25.5 kg·m^2^; (c) at least 3 years of cycling training experience of 6–12 h/week. Subjects were excluded if they: (a) had any metabolic or cardiovascular pathology or abnormality; (b) smoked or drank regularly; (c) took supplements or medication in the previous 2 weeks or had abnormal values in the pre-inclusion blood test; (d) suffered an injury in the last six months. All participants signed the informed consent form. The research was conducted according to the guidelines of the Declaration of Helsinki for Research on Human Subjects [20] and was approved by the Ethics Committee of the University of the Catholic University of Murcia (CE091802).

This descriptive study consisted of 3 laboratory visits. Visit 1: consisted of a medical examination, a blood draw to determine health status, and a familiarization session with the cycloergometer. Visit 2: included a 3-day 24 h diet reminder and an incremental test to exhaustion on a cycle ergometer. Visit 3: occurred after 8 weeks, and another 3-day 24 h diet reminder and another incremental test to exhaustion on a cycle ergometer were performed. During visits 2 and 3, finger capillary blood draws were taken after completion of the incremental test to analyze blood gas markers (ABL90 Flex). In addition, before each testing session (visits 2 and 3), the nutritionist prescribed a standardized breakfast (557.7 kcal) consisting of 95.2 g carbohydrate (68%), 18.9 g protein (14%), and 11.3 g lipids (18%). Cyclists were told to continue with their usual diet and training schedule. In addition, this study was conducted from the end of September to the end of December (off-season), a period when cyclists decrease their volume and intensity of training.

A medical examination included a health history, resting electrocardiogram, and cardiorespiratory examination (auscultation, blood pressure, etc.) and confirmed that the volunteer was healthy to participate in the study.

Two venous blood samples (3.5 mL tube with polyethylene terephthalate (PET) and 3.0 mL tube with ethylenediaminetetraacetic acid (EDTA)) were taken to check the health status of the participants. Erythrocyte counts were performed on a Cell-Dyn 3700 automated analyzer (Abbott Diagnostics, Chicago, IL, USA) using internal (Cell-Dyn 22) and external (Programme of Excellence for Medical Laboratories-PEML) controls. Erythrocyte, haemoglobin, haematocrit, and haemacytometry indices were analyzed.

An incremental test with a final ramp test was performed on a bicycle ergometer (Cyclus 2, RBM Elektronik-Automation GmbH, Leipzig, Germany) using a metabolic cart (Metalyzer 3B. Leipzig, Germany) to determine the maximal fat oxidation zone (FatMax), VT1 and VT2, and VO_2MAX_. Participants started pedaling at 35 W for 2 min, and then the intensity increased by 35 W every 2 min until an RER > 1.05 was reached. Then, participants completed a final ramp of 35 W·min^−1^ until voluntary exhaustion. To ensure the VO_2MAX_, at least 2 of the following criteria had to be reached: plateau in the final VO_2_ values (increase ≤2.0 mL·kg^−1^·min^−1^ in the last 2 loads) and an achieved maximum theoretical heart rate (HR) ((220 − age) − 0.95), RER ≥ 1.15, and lactate ≥ 8.0 mmol·L^−1^. Ventilatory thresholds (VT1 and VT2) were obtained using the ventilatory equivalents method described by Wasserman [21]. FTP was defined as the highest mean power output (PO) that can be maintained for 1 h. The estimated functional threshold power (FTP) was calculated using the following equation [22]:FTP (W) = Pmax (W) × 0.865 − 56.484

The blood parameters were haematocrit (Hct), haemoglobin (Hb), oxygen partial pressure (pO_2_) and carbon dioxide (pCO_2_), oxygen pressure (sO_2_), oxyhaemoglobin (O_2_Hb), carboxy-hemoglobin (COHb), deoxyhemoglobin (RHb), methemoglobin (MetHb), total blood oxygen concentration (tO_2_), total blood carbon dioxide concentration (tCO_2_), oxygen partial pressure at 50% oxygen saturation (p50), non-oxygenated blood fraction (Shunt), and the difference between the alveolar concentration (A) of oxygen and the arterial (a) concentration of the oxygen (AaDpO_2_). The acid-base parameters were pH, lactate (La^−^), bicarbonate (HCO_3_^−^), standard bicarbonate (SBC), actual base excess (ABE), and standard base excess (SBE), and the electrolytes parameters were Na^+^, K^+^, Ca^+^, and Cl^−^, and the difference between anions and cations in plasma (Anion gap) and plasma osmolarity (mOsm) was determined via arterialized capillary blood from the fingertip at rest after the end of the incremental test before and after the off-season period. The ABL 90 FLEX blood gas analyzer (Radiometer Medical ApS, Copenhagen, Denmark) measured the above-mentioned parameters and was calibrated at hourly intervals throughout the day, with internal reference standards. A previous study indicated that ABL90 FLEX has good accuracy. The plastic capillary tubes were preheparinized with electrolytically balanced solid heparin. This significantly reduces the risk of clots and helps to ensure reliable results without electrolyte bias.

The statistical analysis was performed using the Statistical Package for Social Sciences (SPSS 21.0, International Business Machines, Chicago, IL, USA). Descriptive statistics are presented as mean ± standard deviation (SD). Levene and Shapiro–Wilks tests checked for homogeneity and normality of the data, respectively. A Student’s t-test or Wilcoxon for paired data evaluated differences pre-post-intervention. Additionally, the standardized mean differences were calculated using Cohen’s effect size (ES) (95% confidence interval) for all comparisons. Threshold values for ES statistics were >0.2 small, >0.5 moderate, and >0.8 large. The different correlations between the parameters were evaluated using Pearson’s or Spearman’s correlation (r). The significance level was set at *p* ≤ 0.05.

## 3. Results

Table 1 and Table 2 show the data on the characteristics of the well-trained cyclists and the nutritional changes after the off-season period, respectively. Although we observed no significant change in weight, we did find a decrease in fat intake without significant changes in total calorie intake.

In Table 3, we found changes in the markers of oxygen metabolism (blood gas) in capillary blood at the end of the 8-week off-season in well-trained cyclists. Unexpectedly, a significant increase in sO_2_ and O_2_Hb was observed; however, a significant increase in COHb and RHb was also found. In addition, a decrease in Shunt and AaDpO_2_ was found.

In Table 4, we show the values found in markers of acid-base status (blood gasometry) in capillary blood after 8 weeks of the off-season in well-trained cyclists. There were no significant changes in acid-base status. In addition, no changes in electrolyte markers were observed (Table 5).

When evaluating the metabolic data (Table 6), we found a decrease in FatMax fatty acid oxidation (FatMax-FOR) and a decreasing trend in absolute oxygen consumption in the VT2 (VT2-VO_2_) and VO_2MAX_ at the end of the off-season period in well-trained cyclists. However, although a decrease of −1.73% was observed in the time-to-exhaustion in the incremental test, this was not significant.

Table 7 presents the correlations found when comparing the performance and metabolic blood gas markers (oxygen metabolism and acid-base status). A significant positive correlation was found between pre-post changes in oxygen consumption at the FatMax (ΔFatMax-VO_2_) and start–end changes in the ΔsO_2_ (r = 0.502, *p* = 0.03), and a negative correlation was observed with start–end changes in ΔHCO_3_^−^ (r = −0.466, *p* = 0.04) (Table 7). Furthermore, a significant negative correlation was shown between ΔFatMax-FOR and ΔsO_2_ (r = −0.575, *p* = 0.01) and between changes in the FatMax power output (ΔFatMax-W) with ΔsO_2_ (r = −0.689, *p* ≤ 0.01) (Table 7).

In Figure 1, the well-trained cyclists had a positive correlation between the ΔFatMax-W and ΔFatMax-VO_2_ (r = 0.813; R^2^ = 0.661; *p* ≤ 0.01) with ΔFatMax-FOR (r = 0.702; R^2^ = 0.493; *p* ≤ 0.01) and with the time-to-exhaustion in the incremental test (r = 0.607; R^2^ = 0.369; *p* ≤ 0.01). Firstly, these correlations indicate that cyclists who decreased their oxygen consumption the most in the FatMax also decreased their power output the greatest in the FatMax, and secondly, cyclists who decreased their fat oxidation capacity the largest in the FatMax decreased their time duration in the incremental test (lower performance).

On the other hand, a significant negative correlation was found between start–end changes in power generated at the estimated functional threshold power (ΔeFTP-W) and ΔpH (r = −0.546, *p* = 0.02) (Table 7). Furthermore, a negative correlation was observed between start–end changes in power generated in the VO_2MAX_ (ΔVO_2MAX_-W) and ΔpH (r = −0.554, *p* = 0.02) and between time incremental test and ΔsO_2_ (r = −0.601, *p* = 0.01) (Table 7). In Figure 2, the well-trained cyclists had a positive correlation between the ΔVO_2MAX_-W and ΔeFTP-W (r = 0.999; R^2^ = 0.999; *p* ≤ 0.01) and a negative correlation between ΔpH with ΔVO_2MAX_-W (r = −0.554; R^2^ = 0.307; *p* ≤ 0.01) and the ΔeFTP-W (r = −0.546; R^2^ = 0.298; *p* ≤ 0.01). This indicates that cyclists who increased pH the most at the end of the off-season period had a greater decrease of the power output generated in the eFTP and VO_2MAX_.

In Figure 3, the well-trained cyclists had a positive correlation between ΔHCO_3_^−^ con ΔpCO_2_ (r = 0.959, R^2^ = 0.919, *p* ≤ 0.01) and ΔpH (r = 0.702, R^2^ = 0.493, *p* ≤ 0.01). Furthermore, in relation to the ΔLa^−^, there was a significant inverse correlation with the ΔpH and ΔHCO_3_^−^ (r = −0.499, R^2^ = 0.249, *p* = 0.04; r = −0.550, R^2^ = 0.302, and *p* = 0.02, respectively). This indicates that cyclists who increased La^−^ the most had greater decreases in pH and HCO_3_^−^ after the off-season period.

## 4. Discussion

The main objective of this study was to evaluate the effect of an 8 weeks off-season period (↓volume and intensity of training) on capillary blood gas markers (oxygen metabolism and acid-base status) in well-trained cyclists. The main findings were changes in the markers of oxygen metabolism and gas exchange, with no change in acid-base status and electrolytes after the off-season period in well-trained cyclists.

In relation to nutrient intake, there was only a significant decrease in fat (−14.3%; *p* = 0.043); however, in caloric intake, there was no significant difference (1.6%; *p* = 0.729) after the off-season period in amateur cyclists. This suggests that the dietary pattern of the cyclists was very stable in this phase of the season.

After the 8-week off-season period, where well-trained cyclists continued to train but at a lower intensity and volume, we found an increase in sO_2_, O_2_Hb, and COHb, with decreases in RHb, Shunt, and AaDpO_2_. No previous studies have evaluated this biomarker in the off-season. However, it has been observed that after 5 weeks of cessation of training in young top-level road cyclists, there was a significant decrease in red blood cells (−6.6%), Hb (−5.4%), and Hct (−2.9%) [4]. In contrast, we found no change in hemoglobin and hematocrit. These differences may be mainly due to differences in the off-season conditions (the type of exercise, volume, and intensity) between the two studies. Therefore, according to the data from our study and taking into account the volume and intensity performed in the off-season, well-trained cyclists are not negatively affected by the markers of oxygen metabolism.

We found no significant changes within the acid-base status markers. In line with our results, Lucia et al. [2] also did not observe significant changes in HCO_3_^−^ and pH between the off-season, pre-season, and season during an incremental test in professional cyclists. Although their protocol for assessing changes in HCO_3_^−^ and pH was different from our study (a rectangular test), it appears that the degree of training does not affect these acid-base status markers. However, the same author observed a decrease in lactate across the different periods of the cycling season (the off-season, pre-season, and competition) in an incremental test [2]. This suggests that there is an increase in blood lactate from the end of the competition period and the beginning of the off-season to pre-season, possibly due to a reduced capacity for fatty acid oxidation coupled with mitochondrial decrease [23]. It should be noted that one of the main attributes of professional road cyclists is their ability to sustain high absolute work rates for prolonged periods of time while maintaining stable blood lactate concentrations [24]. Therefore, a decrease in this capacity would negatively affect performance at high cycling intensities. In line with this affirmation are the correlations found between Lac with pH and HCO_3_^−^, where well-trained cyclists who had greater increases in Lac had greater decreases in pH and HCO_3_^−^, indicating higher metabolic stress in these cyclists after the off-season period because of more acidification of the blood.

In addition, another study that measured the acid-base status in a tapering period (a 53% decrease in training volume over 14 days) in swimmers observed no significant differences in pH, HCO_3_^−^, and ABE after a test of 182.9 m [25]. These results agree with ours, as we also found no change in these parameters after an incremental test, which may be explained by no change in blood lactate either. Overall, a decrease in training volume and intensity does not seem to affect the body’s buffer system. Therefore, the measurement of acid-base status and metabolic parameters (such as pH, HCO_3_^−^, ABE, and Lac) in the off-season period would be interesting to analyze in order to know the magnitude of changes and to be able to help coaches to establish pre-season training based on a complete physiological profile.

After the off-season period, we found a decrease in FatMax-FOR and a tendency towards a decrease in the VT2-VO_2_ and VO_2MAX_. However, Sassi et al. [3] found a significant increase in absolute peak power and relative to weight and power in the VT2, no change in the VO_2_ at the VO_2MAX_ and VT2 in the absolute and relative to weight values from the off-season to pre-season (3–4 weeks active rest and 8–15 days of easy training). Similar to our results, Coyle et al. [7] found a decrease in the VO_2MAX_ at 21 days without any physical activity, stabilizing at 56 days in trained cyclists, possibly due to a decrease in maximum stroke volume and maximum arterio-venous oxygen difference. The same author also found changes at the muscle fiber level, specifically a decrease in citrate synthase and succinate dehydrogenase. On the other hand, the cessation of training (five weeks) in young, trained cyclists caused a −8.8% in the VO_2MAX_, −6.5% in the peak power output, −12.9% in the VT1 power output, and −11.5% in the VT2 [4]. Decreasing training volume and intensity in endurance athletes may decrease the VO_2MAX_ due to a reduction in cardiac dimensions and ventilatory efficiency [6,26], also affected by a lower oxygen supply to the muscle [27]. Therefore, the degree of decrease in performance, as measured by exercise time- to-fatigue and the VO_2MAX_, is likely influenced by the mode of exercise, intensity, and duration of detraining.

In the FatMax exercise zone, decreased the ΔFatMax-VO_2_ correlated with reduced fat oxidation at the FatMax (ΔFatMax-FOR), power output at the FatMax (ΔFatMax-W), and time-to- exhaustion in the incremental test (ΔTIME). The findings indicate that metabolic changes in the FatMax (↓VO_2_) at low–moderate exercise intensities are associated with decreased performance in both these exercise zones and time-to-exhaustion in maximal tests after an off-season period in well-trained cyclists. In addition, we also observed an inverse correlation between pH with the eFTP-W and the VO_2MAX_-W, implying that well-trained cyclists who had decreased pH the most had a greater increase in the watts generated in the eFTP and VO_2MAX_ after the off-season period. Knowledge of the metabolic and performance changes mediated by detraining provides valuable information for coaches, as they can establish a more targeted training programme based on these changes. In addition, sports scientists can increase their knowledge of the effects of detraining on cyclists.

A recently published study that evaluated the changes at the physiological and performance level observed no changes in the absolute and relative VO_2MAX_ after a period where the load was decreased between 5 and 25% due to COVID-19 restrictions (30 days) in U23 cyclists [28]. In contrast, another study observed a significant decrease in training characteristics (↓33.9% of the total training volume), with losses between 1% and 19% in the best 5-min and best 20-min performance in all professional cyclists analyzed during seven weeks of COVID-19 confinement [29]. The differences observed between the two studies may be because professional athletes coped better with COVID-19 confinement than well-trained athletes, although both cohorts of cyclists reported reductions in training volume, intensity, and frequency. Another reason for the differences between studies could be due to the different COVID-19 restrictions between countries [30]. As can be seen from the data presented in this discussion, it is important to know the characteristics of detraining (off-season and COVID-19 confinement) in order to understand the physiological–biochemical changes that may occur.

## 5. Conclusions

After the off-season period, the well-trained cyclists increased markers of oxygen metabolism (sO_2_, O_2_Hb, and COHb), with no change in acid-base status and electrolytes. However, we did find a decrease in fat oxidation in the FatMax and a decreasing trend in the VO_2_ in the VT2 and VO_2MAX_. In addition, we found a relationship between changes in power generated in the eFTP and VO_2MAX_ with changes in the pH. Furthermore, we also found a relationship between changes in the pH and HCO_3_^−^ with changes in the La^−^.

### Practical Applications

Monitoring markers of oxygen metabolism, acid-base status, and performance after an off-season period is key to understanding the global physiological changes that occur during the entire cycling season [31,32,33]. These findings provide valuable insight for coaches as to what happens physiologically after the off-season period so that they can plan training sessions appropriately.

## Figures and Tables

**Figure 1 nutrients-14-03808-f001:**
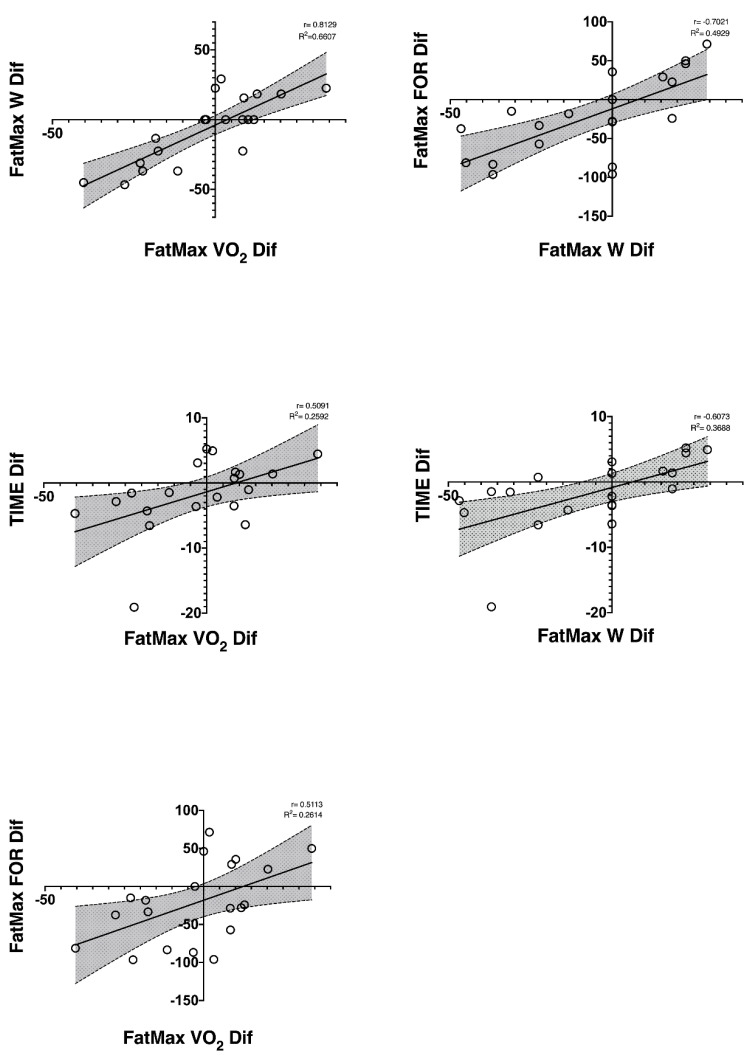
Correlations found between metabolic and performance variables in the FatMax after an incremental test at the end of the off-season period in well-trained cyclists. All correlations were significant <0.05.

**Figure 2 nutrients-14-03808-f002:**
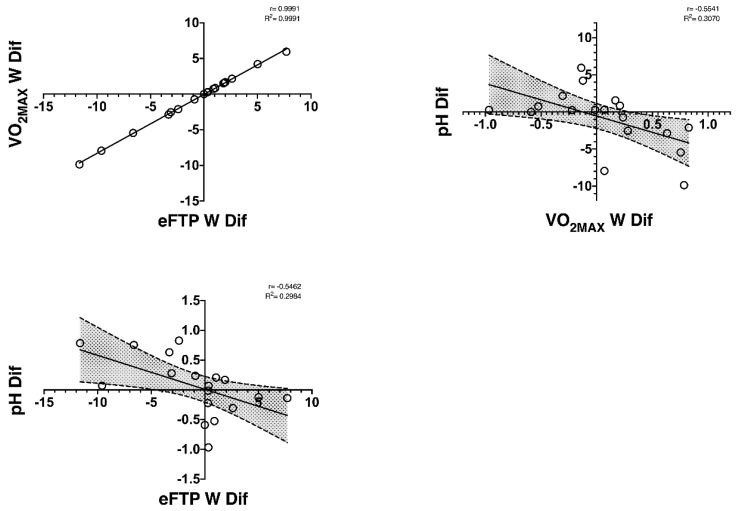
Correlations found between performance variables (the VO_2MAX_ W and eFTP) and capillary blood gases after an incremental test at the end of the off-season period in well-trained cyclists. All correlations were significant <0.05.

**Figure 3 nutrients-14-03808-f003:**
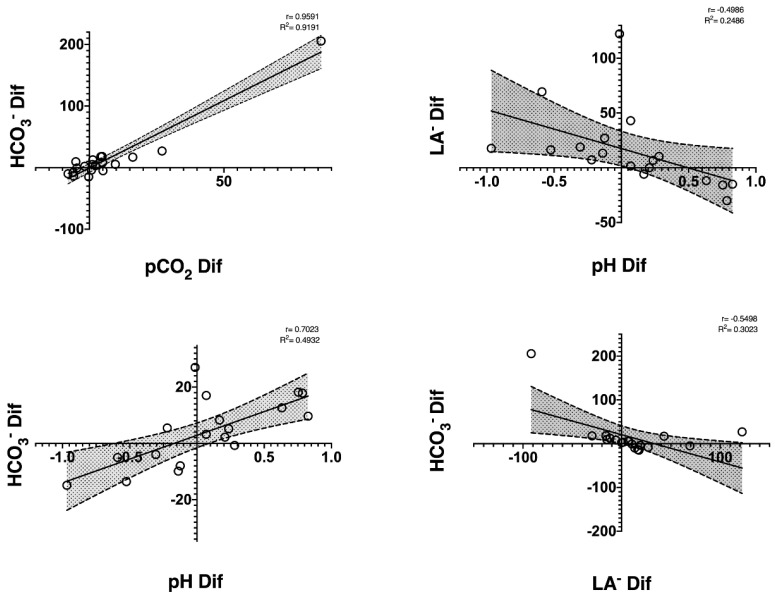
Correlations found between capillary blood gases variables after an incremental test at the end of the off-season period in well-trained cyclists. All correlations were significant at <0.05.

**Table 1 nutrients-14-03808-t001:** Baseline general characteristics and training variables of well-trained cyclists during the off-season period.

	Start	End	*p*-Value
Age (years)	32.6 (9.12)		
Height (cm)	177.0 (6.25)		
Body mass (kg)	70.4 (6.06)	70.7 (6.15)	0.374
BF (%)	9.0 (1.64)	9.0 (1.63)	0.775
Training variables	**During the off-season period**		
Total distance (km)	1082 (810.0)		
HR_AVG_ (bpm)	137 (13.1)		
W_AVG_ (W)	163.5 (32.5)		
RPE	6.3 (1.2)		
Zone 1 (%)	26.7 (17.8)		
Zone 2 (%)	24.0 (7.31)		
Zone 3 (%)	20.3 (10.1)		
Zone 4 (%)	21.7 (14.2)		
Zone 5 (%)	7.3 (6.0)		

Values are expressed as mean (SD). BF % = body fat percentage; BMI = body mass index; HRavg = average heart rate of all the training sessions carried out during the study period; Total distance = all the training sessions carried out during the study period; RPE = rating perceived of exertion; Wavg = average power output of all training sessions during the study period.

**Table 2 nutrients-14-03808-t002:** Nutritional comparison at the start and end of the off-season period of well-trained cyclists.

	Start	End	*p*-Value
**Kilocalories**	2100 (529)	2134 (449)	0.729
**Carbohydrates** (g)	222 (71.5)	248 (59.7)	0.084
**Lipids** (g)	84 (24.3)	72 (19.4)	**0.043 ***
**Proteins** (g)	115 (26.0)	102 (24.3)	0.177

Values are expressed as mean (SD). The mean values correspond to the average of all 24-h diet recall data collected at pre-intervention (three days) and post-intervention (three days). * Indicates significant differences (*p* ≤ 0.05).

**Table 3 nutrients-14-03808-t003:** Changes in oxygen metabolism markers (ABL 90) during the off-season in well-trained cyclists.

	Start	End	*p*-Value	ES	∆%
**Hct** (%)	49.4 (3.22)	49.3 (2.47)	0.980	0.01	0.19
**Hb** (g/dL)	16.1 (1.05)	16.1 (0.80)	0.939	0.20	0.12
**pCO_2_** (mmHg)	34.1 (4.26)	36.0 (3.12)	0.295	0.28	7.0
**pO_2_** (mmHg)	0.29 (0.90)	0.26 (0.14)	0.276	0.27	3.6
**sO_2_** (%)	94.2 (1.36)	95.1 (1.24)	**0.024 ***	0.56	0.96
**O_2_Hb** (%)	93.1 (1.34)	93.9 (1.23)	**0.041 ***	0.51	0.86
**COHb** (%)	0.461 (0.11)	0.505 (0.17)	**0.026 ***	0.63	27.7
**RHb** (%)	5.67 (1.35)	4.79 (1.22)	**0.023 ***	0.57	−12.3
**MetHb** (%)	0.789 (0.11)	0.795 (0.12)	0.971	0.02	0.90
**tO_2_** (mmol/L)	9.43 (0.60)	9.50 (0.51)	0.497	0.16	−4.1
**tCO_2_** (mmol/L)	14.6 (2.54)	16.0 (3.33)	0.205	0.34	−13.9
**p50** (mmHg)	31.3 (2.96)	30.1 (2.83)	0.181	0.37	−3.2
**Shunt** (%)	12.6 (3.86)	9.79 (3.14)	**0.021 ***	0.60	−16.1
**AaDpO_2_** (mmHg)	25.9 (6.11)	22.2 (4.68)	**0.032 ***	0.55	−10.4

Values are mean (SD). AaDpO_2_ = difference between the alveolar concentration (A) of oxygen and the arterial (a) concentration of oxygen; COHb = carboxy-hemoglobin; Hb = hemoglobin; Hct = haematocrit; MetHb = methemoglobin; O_2_Hb = oxyhemoglobin; RHb = deoxyhemoglobin; p50 = oxygen partial pressure at 50% oxygen saturation; pCO_2_ = carbon dioxide partial pressure; pO_2_ = oxygen partial pressure; Shunt = non-oxygenated blood fraction; sO_2_ = oxygen saturation; tCO_2_ = total blood carbon dioxide concentration; tO_2_ = total blood oxygen concentration. * Indicates significant differences (*p* ≤ 0.05).

**Table 4 nutrients-14-03808-t004:** Changes in acid-base status markers (ABL 90) during the off-season period in well-trained cyclists.

	Start	End	*p*-Value	ES	∆%
**pH**	7.200 (0.05)	7.210 (0.03)	0.607	0.12	0.06
**HCO_3_^−^** (mmol/L)	13.5 (2.43)	14.9 (3.25)	0.241	0.32	14.5
**SBC** (mmol/L)	14.5 (1.77)	15.5 (2.75)	0.277	0.29	8.0
**ABE** (mmol/L)	−13.4 (2.84)	−11.9 (3.75)	0.251	0.31	−6.3
**SBE** (mmol/L)	−14.5 (3.05)	−12.8 (4.07)	0.226	0.29	−6.5
**La^−^** (mmol/L)	14.4 (3.50)	14.8 (3.64)	0.660	0.10	9.7

Values are mean (SD). ABE = actual base excess; HCO_3_^−^ = bicarbonate anion; La^−^ = lactate; SBC = standard bicarbonate; SBE = standard base excess.

**Table 5 nutrients-14-03808-t005:** Changes in electrolyte markers (ABL 90) during the off-season in well-trained cyclists.

	Start	End	*p*-Value	ES	∆%
**Na^+^** (mmol/L)	146 (1.65)	146 (2.06)	0.881	0.04	−0.03
**K^+^** (mmol/L)	5.56 (1.60)	4.94 (0.45)	0.477	0.20	−5.7
**Ca^+^** (mmol/L)	1.31 (0.08)	1.28 (0.03)	0.176	0.36	−2.0
**Cl^−^** (mmol/L)	110 (3.02)	109 (2.26)	0.392	0.27	−0.80
**Anion gap** (mmol/L)	22.1 (3.66)	22.2 (3.89)	0.636	0.14	6.7
**mOsm** (mmol/kg)	298 (3.66)	298 (4.35)	0.740	0.10	−0.01

Values are mean (SD). Anion gap = difference between anions and cations in plasma and mOsm = plasma osmolarity.

**Table 6 nutrients-14-03808-t006:** Changes in metabolic and performance markers during the off-season in well-trained cyclists.

	Start	End	*p*-Value	ES	∆%
**FatMax HR** (ppm)	131 (13.70)	127.7 (12.0)	0.137	0.36	−3.4
**FatMax VO_2_** (L·min^−1^)	2.1 (0.50)	2.0 (0.58)	0.494	0.16	−2.4
**FatMax VO_2MAX_** (%)	52.5 (7.59)	52.4 (9.47)	0.962	0.01	0.67
**FatMax FOR** (g·min^−1^)	20.1 (8.19)	15.3 (10.6)	**0.026 ***	0.54	−21.5
**FatMax W** (watts)	173 (38.50)	159 (46.70)	0.153	0.43	−6.4
**VT1 HR** (ppm)	128 (10.90)	129 (10.10)	0.699	0.09	1.2
**VT1 VO_2_** (L·min^−1^)	1.99 (0.40)	2.05 (0.48)	0.369	0.21	3.3
**VT1 VO_2MAX_** (%)	50.0 (4.78)	52.9 (5.39)	0.075	0.42	6.7
**VT1 W** (watts)	153 (33.10)	159 (37.50)	0.453	0.17	0.90
**eFTP** (watts)	279 (45.6)	276 (42.9)	0.388	0.20	−0.67
**VT2 HR** (ppm)	168 (10.90)	166 (8.13)	0.417	0.19	−0.75
**VT2 VO_2_** (L·min^−1^)	3.35 (0.56)	3.21 (0.57)	**0.055 ^#^**	0.34	−3.9
**VT2 VO_2MAX_** (%)	84.0 (5.70)	82.8 (5.56)	0.399	0.19	−1.1
**VT2 W** (watts)	286 (49.90)	277 (53.80)	0.264	0.26	−2.6
**VO_2MAX_ HR** (ppm)	183 (8.68)	183 (7.07)	0.641	0.11	−0.21
**VO_2MAX_** (L·min^−1^)	3.98 (0.63)	3.86 (0.60)	**0.057 ^#^**	0.45	−2.7
**VO_2MAX_** (mL·kg^−1^·min^−1^)	57.9 (9.53)	56.2 (8.42)	0.097	0.39	−2.5
**VO_2MAX_ W** (watts)	388 (52.7)	385 (49.6)	0.888	0.04	−0.57
**Time incremental test** (s)	1247 (168.00)	1225 (177.00)	0.185	0.34	−1.73

Values are mean (SD). eFTP = estimated functional threshold power; FatMax = zone where maximum fat oxidation occurs; FOR = fat oxidation rate; HR = heart rate; VO_2_ = volume of oxygen uptake; VO_2MAX_ = maximum oxygen uptake; VT1 = ventilatory threshold 1; VT2 = ventilatory threshold 2. * Indicates significant differences (*p* ≤ 0.05). ^#^ Indicates tendency (*p* = 0.05–0.06).

**Table 7 nutrients-14-03808-t007:** Correlations of the start–end off-season differences between performance-metabolic variables and blood gas markers in well-trained cyclists.

		ΔsO_2_	ΔpO_2_	ΔpCO_2_	ΔAaDpO_2_	ΔpH	ΔHCO_3_^−^	ΔLa^−^	ΔK^+^	ΔAnion Gap
**ΔFatMax VO_2_** (L·min^−1^)	*r*	0.502	−0.131	−0.423	0.313	0.142	−0.466	0.240	0.310	0.028
	*p-value*	** 0.03 * **	0.60	** 0.07 ^#^ **	0.21	0.58	** 0.04 * **	0.32	0.20	0.92
**ΔFatMax VO_2MAX_** (%)	*r*	−0.372	−0.081	−0.370	0.200	0.264	−0.396	0.129	0.286	−0.049
	*p-value*	0.12	0.75	0.12	0.43	0.29	0.09	0.60	0.24	0.85
**ΔFatMax FOR** (g·min^−1^)	*r*	−0.575	−0.123	−0.395	0.306	−0.145	−0.347	−0.099	0.420	0.356
	*p-value*	** 0.01 * **	0.63	0.10	0.22	0.57	0.15	0.69	**0.07 ^#^**	0.16
**ΔFatMax W** (watts)	*r*	−0.689	−0.244	−0.324	0.393	0.053	−0.363	0.236	0.189	0.333
	*p-value*	** <0.01 * **	0.33	0.18	0.11	0.83	0.13	0.33	0.44	0.19
**ΔVT1 VO_2_** (L·min^−1^)	*r*	−0.212	−0.239	−0.196	0.340	0.003	−0.212	−0.025	0.272	−0.017
	*p-value*	0.38	0.34	0.42	0.17	1.00	0.38	0.92	0.26	0.66
**ΔVT1 VO_2MAX_** (%)	*r*	−0.043	−0.136	−0.150	0.174	0.067	−0.139	−0.124	0.275	−0.221
	*p-value*	0.86	0.59	0.54	0.49	0.79	0.57	0.61	0.26	0.39
**ΔVT1 W** (watts)	*r*	−0.339	−0.291	−0.083	0.277	−0.143	−0.099	0.004	0.260	−0.037
	*p-value*	0.16	0.24	0.74	0.27	0.57	0.69	1.00	0.28	0.89
**ΔeFTP W** (watts)	*r*	−0.104	−0.100	0.063	0.098	−0.546	0.015	0.231	−0.026	0.207
	*p-value*	0.67	0.69	0.80	0.70	**0.02 ***	0.95	0.34	0.91	0.43
**ΔVT2 VO_2_** (L·min^−1^)	*r*	−0.546	−0.241	−0.171	0.263	−0.071	−0.192	0.020	0.191	−0.109
	*p-value*	0.02	0.34	0.48	0.29	0.78	0.43	0.94	0.43	0.68
**ΔVT2 VO_2MAX_** (%)	*r*	−0.305	−0.129	−0.060	−0.003	0.159	−0.037	−0.223	0.160	−0.313
	*p-value*	0.20	0.61	0.81	1.00	0.53	0.88	0.36	0.51	0.22
**ΔVT2 W** (watts)	*r*	−0.489	−0.243	−0.140	0.181	−0.107	−0.174	−0.049	0.216	−0.189
	*p-value*	0.03	0.33	0.57	0.47	0.67	0.48	0.844	0.37	0.47
**VO_2MAX_** (L·min^−1^)	*r*	−0.345	−0.155	−0.139	0.319	−0.255	−0.187	0.270	0.051	0.172
	*p-value*	0.15	0.54	0.57	0.20	0.31	0.44	0.26	0.84	0.51
**VO_2MAX_** (mL·kg^−1^·min^−1^)	*r*	−0.285	−0.112	−0.177	0.287	−0.263	−0.226	0.316	0.059	0.103
	*p-value*	0.24	0.66	0.47	0.25	0.29	0.35	0.19	0.81	0.69
**VO_2MAX_ W** (watts)	*r*	−0.099	−0.101	0.070	0.098	−0.554	0.019	0.235	−0.035	0.213
	*p-value*	0.69	0.69	0.78	0.70	**0.02 ***	0.94	0.33	0.89	0.41
**Time incremental test** (s)	*r*	−0.601	−0.247	−0.151	0.276	−0.123	−0.156	−0.003	0.193	0.363
	*p-value*	**0.01 ***	0.32	0.54	0.27	0.63	0.52	1.00	0.43	0.15

Values are mean (SD). ABE = actual base excess; eFTP = estimated functional threshold power; FatMax = zone where maximum fat oxidation occurs; FOR = fat oxidation rate; HCO_3_^−^ = bicarbonate anion; HR = heart rate; La^−^ = lactate; SBC = standard bicarbonate; SBE = standard base excess VO_2_ = volume of oxygen uptake; VO_2MAX_ = maximum oxygen uptake; VT1 = ventilatory threshold 1; VT2 = ventilatory threshold 2. * Indicates significant differences (*p* ≤ 0.05). ^#^ Indicates tendency (*p* = 0.05–0.07).

## Data Availability

The data presented in this study are available on request from the last author.

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
