# Peer review of "Biomarker Changes in Oxygen Metabolism, Acid-Base Status, and Performance after the Off-Season in Well-Trained Cyclists"

_nutrients, 2022, doi:10.3390/nu14183808_

Round 1

Reviewer 1 Report

Martínez-Noguera et al. describe the biomarker alteration of well-trained cyclists during the off-season. Specifically, they identified increased oxygen metabolism and reduced fat oxidation during the FatMax phase and decreased VO2 during incremental tests after the off-season. The paper is well-structured and well-written. Given the potential of using these biomarkers to predict and guide the recovery of cyclists after the off-season, the authors' research can be important for audiences in the field. Before suggesting publication, I have the following issues that should be resolved.

Major Issues:

1.     The authors have done a great job introducing the background and describing the result. However, this article generally does not emphasize the significance of this research (maybe a little in the practical applications section). It is suggested to put extra effort at least in the end of the introduction and throughout the discussion to emphasize the significance of this research to the field. As the authors mentioned that blood gas markers are rarely measured after detraining, maybe trying to emphasize the advantages, e.g. convenience, of using blood gas markers would be a good direction. Also, since some of the biomarkers are associated with performance, discussing the potential advantages of using these biomarkers to predict performance is also an interesting aspect.

2.     There are too many abbreviates and detailed readouts in the abstract part. This makes the general audience difficult to follow. It is suggested to summarize the results into short sentences and include the whole name of abbreviates the first time they appear. Also, try to add a sentence at the end to emphasize the significance of this work.

Minor Issues:

1.      The format of the table legends and figure legends is not always consistent. Some of them are bold (Table 3, 4, 5) while the other are not (Table 1, 2). Please be consistent about these details.

2.     In line 34, it is “4 weeks” instead of “4 week”.

3.     The title of “Practical Application” is not bold while others are.

Author Response

Martínez-Noguera et al. describe the biomarker alteration of well-trained cyclists during the off-season. Specifically, they identified increased oxygen metabolism and reduced fat oxidation during the FatMax phase and decreased VO2 during incremental tests after the off-season. The paper is well-structured and well-written. Given the potential of using these biomarkers to predict and guide the recovery of cyclists after the off-season, the authors' research can be important for audiences in the field. Before suggesting publication, I have the following issues that should be resolved.

Response: We thank the reviewer for their constructive and helpful feedback on our manuscript. We have replied to each specific comment in the section below and have introduced the corresponding edits into the manuscript using Word’s track changes.

Major Issues:

  1. The authors have done a great job introducing the background and describing the result. However, this article generally does not emphasize the significance of this research (maybe a little in the practical applications section). It is suggested to put extra effort at least in the end of the introduction and throughout the discussion to emphasize the significance of this research to the field. As the authors mentioned that blood gas markers are rarely measured after detraining, maybe trying to emphasize the advantages, e.g. convenience, of using blood gas markers would be a good direction. Also, since some of the biomarkers are associated with performance, discussing the potential advantages of using these biomarkers to predict performance is also an interesting aspect.

Response: Following your suggestion, we have modified the introduction, and in the discussion, we included practical applications of the results of this study.

  1. There are too many abbreviates and detailed readouts in the abstract part. This makes the general audience difficult to follow. It is suggested to summarize the results into short sentences and include the whole name of abbreviates the first time they appear. Also, try to add a sentence at the end to emphasize the significance of this work.

Response: Following your suggestion, we have described the abbreviations and shortened the sentences in the results section of the abstract.

Minor Issues:

  1. The format of the table legends and figure legends is not always consistent. Some of them are bold (Table 3, 4, 5) while the other are not (Table 1, 2). Please be consistent about these details.

Response: Thank you very much for your comment, we have revised the format of the tables.

  1. In line 34, it is “4 weeks” instead of “4 week”.

Response: Amended.

  1. The title of “Practical Application” is not bold while others are.

Response:Amended.

Author comment: We appreciate all the comments made on our manuscript, which helped improve it’s quality.

Reviewer 2 Report

Abstract is not clear, it is difficult understand.. E.g. the relationships / correlations coefficients  should be written open. Is the correlation positive or negative. Many short wording e.g. FatMax.

What is the role of nutrients on the results in this study? Should be discussed?

Author Response

We thank the reviewer for their constructive and helpful feedback on our manuscript. We have replied to each specific comment in the section below and have introduced the corresponding edits into the manuscript using Word’s track changes.

Comments and Suggestions for Authors

Abstract is not clear, it is difficult understand.. E.g. the relationships / correlations coefficients  should be written open. Is the correlation positive or negative. Many short wording e.g. FatMax.

Response: Following your suggestion, we have modified the abstract. However, if you feel that we did not make it clearer, please let us know how we can revise it better.

What is the role of nutrients on the results in this study? Should be discussed?

Response: Following your suggestion, we have introduced a phrase related to nutrition in the discussion. Lines 280-283.

Author comment: We appreciate all the comments made on our manuscript, which helped improve it’s quality.
